# Developing Medication Review Competency in Undergraduate Pharmacy Training: A Self-Assessment by Third-Year Students

**DOI:** 10.3390/ijerph20065079

**Published:** 2023-03-14

**Authors:** Aleksi Westerholm, Katja Leiman, Annika Kiiski, Marika Pohjanoksa-Mäntylä, Anita Mistry, Marja Airaksinen

**Affiliations:** 1Clinical Pharmacy Group, Department of Pharmacology and Pharmacotherapy, Faculty of Pharmacy, University of Helsinki, Viikinkaari 5 E, P.O. Box 56, 00014 Helsinki, Finland; 2Faculty of Pharmacy, Pharmacy and Bank Building, Camperdown/Darlington Campus, University of Sydney, Darlington, NSW 2050, Australia

**Keywords:** medication review, clinical pharmacy, pharmacy education, curriculum development, pharmacy student, self-assessment

## Abstract

Pharmacists are increasingly involved in medication history taking, medication reconciliation, and review in their daily practice. The objectives of this study were to investigate third-year pharmacy students’ self-assessed competency in medication reviews and gather their feedback for further development of medication review training in their curriculum. The study was conducted as a self-assessment of third-year pharmacy students at the completion of their second three-month internship period in a community pharmacy in 2017–2018. The students were assigned to review medications of a real patient under the supervision of a medication review accredited pharmacist during their internship. The self-assessment was carried out via an e-form, which was created for this study. Recently established national medication review competence recommendations for pharmacists were used as a reference. Students (*n* = 95, participation rate: 93%) self-assessed their competency as good or very good in 91% (*n* = 28) of the competency areas listed in the self-assessment. The highest proportion of competencies that were self-assessed as good or very good included using medication risk management databases and evaluating the clinical importance of the information (97%, *n* = 92). The lowest proportion of competencies was found in applying clinical information from the key laboratory tests to patient care and knowing which laboratory tests are most important to monitor in each condition and medication (36%, *n* = 34). The students suggested that their pharmacy education should contain more medication review assignments as group work and that an elective course on medication reviews should be compulsory for all pharmacy students.

## 1. Introduction

Medication review practices are evolving in many countries as pharmacists become increasingly involved in medication history taking, medication reconciliation, and medication review in their daily practice [1,2,3,4,5,6]. Medication reviews have the potential to reduce the risk of medication-related problems, inappropriate drug use, and medication costs in patients who are taking multiple medications, also known as polypharmacy [7,8,9].

In Finland, comprehensive medication review (CMR) was the first collaborative procedure involving pharmacists in solving medication-related problems established in 2005 [2,3,10]. Since then, the procedures have evolved and diversified as patients in different healthcare settings require alternate approaches for reviewing their medications, considering their clinical condition [4,9].

The evolution from medication counseling to more sophisticated pharmaceutical services, such as medication reviews, requires more clinical competency from pharmacists [11]. Figure 1 illustrates the change from a drug-centered approach to a patient-centered approach and the increased level of competency required to conduct medication reviews [12]. As medication reviews are becoming a routine practice for pharmacists, the need for competency criteria has become evident [13,14,15,16,17,18]. In Finland, criteria were established in 2017 [19], basing them on three levels of comprehensiveness of reviews as suggested by Clyne et al. [20]: prescription review, medication review, and comprehensive medication review (Figure 1, Appendix A, [20]). The reviews differ from each other concerning the purpose of the review, resources, and competency needed, use of patient information and records, which medicines are included (only prescription medicines or also over-the-counter medicines), and whether only the list of medicines or also the use of medicines is reviewed. Prescription review is a quick check of the medication list and is included in the routine medication dispensing process. Prescription review includes technical and therapeutic issues related to the medication list to the extent that they can be resolved based on the information available in the prescriptions, e.g., dosage and indication. Medication review can be considered a separate service provided to the patient, where the appropriateness of medication is reviewed and therapeutically significant medication-related problems are identified and resolved. Medication review is recommended to be conducted in an interprofessional collaboration, and the key medication observations and changes are discussed with the patient. Medication review includes ensuring appropriate medication use, adherence, and self-management. Comprehensive medication review (CMR) is a more extensive review compared to the medication review. Clinically significant problems related to the medication and health status will be resolved in collaboration with the attending physician and other care team members. Key medication observations and changes are discussed with the patient. CMR includes ensuring the appropriateness of each medication, taking into consideration the patient’s disease and medical condition. A national committee consisting of the key stakeholders responsible for community pharmacy practice development coordinates the establishment of the medication review criteria and competency accreditation [19]. These key stakeholders include competent authorities, universities responsible for pharmacy education, organizations providing continuing education and specialization training, and community pharmacy advocacy organizations.

In Finland, practicing pharmacists can achieve medication review competency by completing a continuing education module of 20 ECTS credits (European Credit Transfer System, 1 credit corresponds to 27 h of student work) or through a portfolio procedure. The training is nationally coordinated [21]. For the comprehensive medication review accreditation, an additional training of 15 ECTS credits is required [2,21]. The increasing need for medication reviews [22], particularly for older adults, has caused a growing demand for competent pharmacists to conduct them. In this context, the medication review competencies were added to the learning outcomes of the undergraduate pharmacy education (BSc curriculum, i.e., the first three years of studies according to the Bologna process [23]). This was implemented at the University of Helsinki, Faculty of Pharmacy, in 2014 as part of a curriculum reform. The first students with medication review competency graduated in 2017. The university education program for pharmacists has two tiers in Finland: BSc (3 years, 180 ECTS credits) and MSc (5 years, 180 + 120 ECTS credits) curricula [24,25]. Since the curriculum reform in 2014, medication review competence has been integrated throughout the BSc (Pharm) curriculum in different study modules and courses, especially in clinical and social pharmacy, pharmacology, biopharmacy, patient counseling and applied pharmacotherapy, rational and safe pharmacotherapy, and internships. Various teaching methods such as interactive lectures, workshops, group work, real patient cases and paper cases, role plays, and seminars with student presentations are utilized.

Although medication reviews are a timely and priority development area of pharmacy practice in diversified inpatient and outpatient care settings throughout the world [1,2,3,4,5,26,27], little research has focused on investigating how required competency is fostered in undergraduate pharmacy education. The objective of this study was to assess third-year pharmacy students’ competency in medication reviews and their feedback for further development of medication review training in their curriculum.

## 2. Methods

The study was conducted as a self-assessment among third-year pharmacy students of the University of Helsinki completing their second period of the required community pharmacy internship in November 2017–January 2018 (the last 3 months out of 6 months). The students were assigned to review the medications of a real patient under the supervision of a medication review accredited pharmacist or a physician. The medication review took place in a community pharmacy or at the patient’s residence. The patient was selected either by the student or an accredited pharmacist. There were no strict selection criteria for the patient: the most important thing was that the student found a suitable patient and practiced reviewing the medication. Patients who had multiple medications in use and wanted to take part in a medication review conducted by a student were suitable.

The medication review included interviewing the patient, analyzing the patient data and medication list, identifying significant medication-related problems, and making evidence-based strategies to solve them. The students were instructed to report their findings to the attending physician.

After completing the required medication review assignment, students self-assessed their medication review competency by using an electronic self-assessment instrument. The self-assessment instrument was based on the national medication review competency criteria for pharmacists established in Finland in 2017 [19,28]. The criteria include recommended competence areas for prescription reviews (17 competence items), medication reviews (prescription review competence items and 11 additional ones), and comprehensive medication reviews (3 additional items, [28]). The competence criteria take into account legislation, the Finnish Medicines Agency’s guidelines on optimizing pharmacotherapy of older adults, Medicines Policy 2020 by the Ministry of Social Affairs and Health, national and international research on collaborative medication review practices (particularly references: [2,4,10,20]), and current undergraduate and continuing education on clinical pharmacy and medication reviews.

Students were instructed to self-assess their competency in both prescription reviews and medication reviews. The competence criteria for these two types of medication reviews [19] were listed in the self-assessment instrument. A 5-point Likert scale, ranging from 1 (very poor/not at all) to 5 (very good), was used as the self-assessment tool for each statement.

Students were asked to disclose the commencement year of their studies in pharmacy, whether they were on a BSc or MSc curriculum, had previous academic, professional, or vocational degrees, or had taken an elective course “Comprehensive medication review and clinical pharmacy” (4 ECTS credits at the time of the study). The grades earned from the three required core courses that provided the foundational knowledge needed to achieve pharmacotherapy and medication review competency (Pathology and Nutrition, Systematic Pharmacology, Medication Counselling, and Pharmacotherapy) were also requested. The self-assessment instrument was pilot tested by students and practicing pharmacists (*n* = 11), and their feedback was considered when structuring its final format.

Data were analyzed for descriptive statistics by Microsoft Excel (version 2016). Descriptive statistics are presented as percentages, means, standard deviations, and a summative scale. The summative variable, competency score, was formed by calculating students’ self-estimates for each of the 28 individual competence items. Each competency score could range from 1 to 5, depending on the student’s self-assessment estimate. Thus, the total competency score range for 28 items was 28–140. Then, competency scores were categorized into 5 grades so that competency scores 28–50 yielded grade 1 (very poor competency); 51–72 yielded grade 2 (poor competency); 73–95 yielded grade 3 (moderate competency); 96–117 yielded grade 4 (good competency); and 118–140 yielded grade 5 (very good competency).

Responses to an open-ended question “How could the teaching of medication reviews be generally developed in the faculty of pharmacy?” were content analyzed. First, all responses to the open-ended question were collected. After that, the responses were categorized by theme. The most common themes were presented in this study. A.W. and K.L. participated in the content analysis of the responses to the open-ended questions.

### Ethical Considerations

The study followed the guidelines of the Finnish National Advisory Board on Research Integrity [29] and Ethical Review Board in the Humanities and Social and Behavioural Sciences, University of Helsinki [30]. Based on their instructions, the study was deemed exempt from requiring formal ethics committee approval. Students were informed that responding to the e-form was regarded as informed consent, and the written cover letter informed them that their responses would be anonymously used for research purposes.

## 3. Results

Altogether, 95 students out of 102 (participation rate: 93%) completed the self-assessment (Table 1). A majority (78%) of them were BSc students and had studied pharmacy on average for 3 years (range 3–6 years). Of the students, 70% (*n* = 66) did not have a previous academic, professional, or vocational degree. Three students had a healthcare-related previous degree.

### 3.1. Prescription Review Competency

A high proportion of students self-estimated their competency as good or very good in most of the 17 competence areas required for conducting prescription reviews (Figure 2). No students estimated their competency as “very poor or not at all” in any of the competence areas. The highest proportion of good or very good competency self-estimates was found in (1) using databases and information systems for prescription reviews and evaluating the clinical importance of the information while considering each patient’s condition (97% of the students had a self-estimate of good or very good), (2) understanding the importance of medication reconciliation and prescription review in improving medication safety and outcomes (95%), and (3) knowing the basic principles of prescribing, dispensing, and reimbursing medicines (95%) (Figure 2). The lowest proportion of good or very good competency self-estimates concerned: mastering means of monitoring and improving medication adherence and self-management (61% good or very good competency); mastering means of prospectively preventing medication-related problems in different patient and therapeutic groups and solving actual problems (63%); and knowing clinical pharmacotherapy and guidelines and how to apply knowledge to patient care to the extent necessary in prescription reviews (74%).

**Figure 2 ijerph-20-05079-f002:**
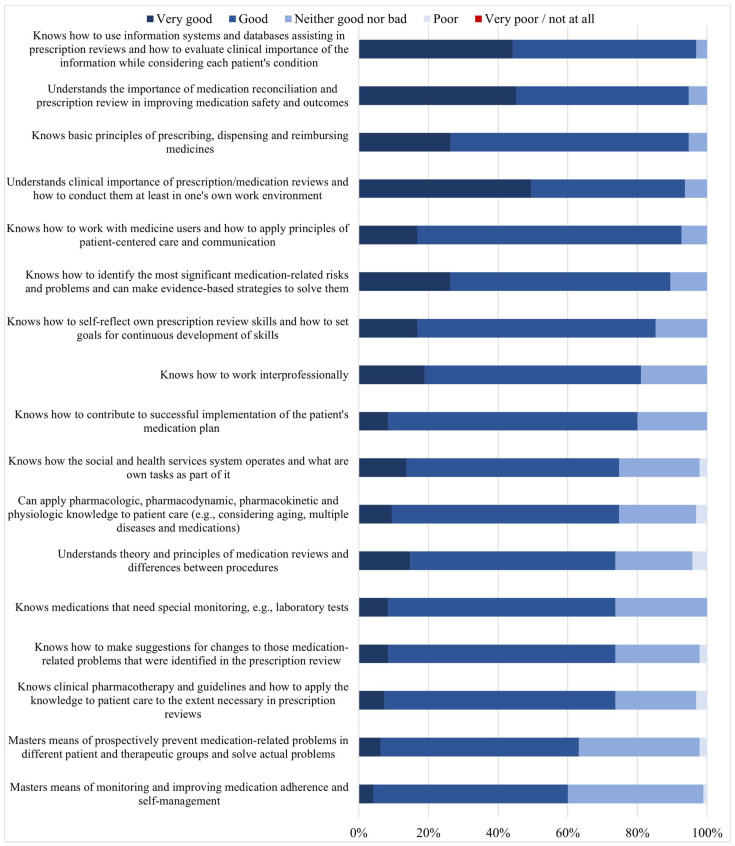
Students’ (*n* = 95) self-assessed competency in conducting prescription reviews (17 competence items, % of the students).

### 3.2. Medication Review Competency

Of the 11 competence items specific to medication reviews, the highest proportion of good or very good competency self-estimates were found for (1) working in a care team and making valid observations on patients’ medications (79% of the students had a self-estimate of a good or very good competency), (2) reviewing the appropriateness of the entire medication regimen of the patient (73%), and (3) evaluating the clinical significance of observations, forming proposals for medication changes and their implementation, and also contributing to their actual implementation (69%) (Figure 3). The lowest proportion of good or very good competency self-estimates concerned: applying clinical information from key laboratory tests to patient care and knowing which laboratory tests are most important to monitor in each condition and medication (36%), knowing how to create contacts with social and health care units (41%), and mastering key principles of implementing medication changes, such as deprescribing and related monitoring of outcomes (55%).

**Figure 3 ijerph-20-05079-f003:**
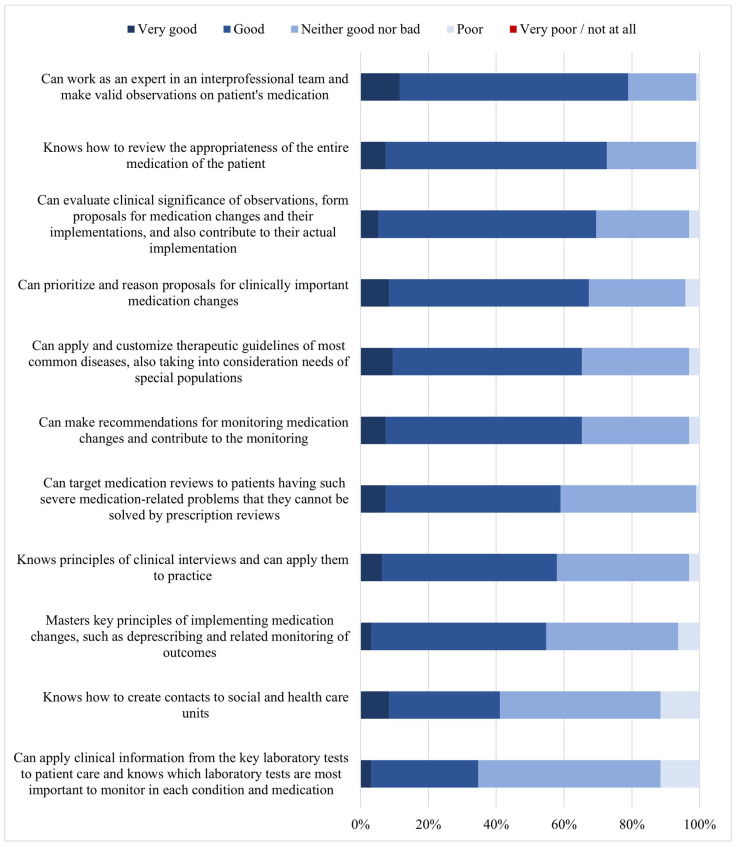
Students’ (*n* = 95) self-assessment of competencies required for medication reviews (*n* = 11) in addition to those required for prescription reviews (*n* = 17, Figure 2) (% of the respondents).

A majority (91%) of students achieved a grade of good (75%) or very good (16%) in summative competency scores for conducting medication reviews. (Figure 4). Moderate was the lowest grade earned based on summative competency scores (9% of the students).

Over half (54%, *n* = 51) of the students had suggestions for further development of medication review training at the University of Helsinki. Students suggested in open-ended comments that their education should contain more medication review assignments and patient cases as group work (*n* = 35). Furthermore, they suggested that the elective “Comprehensive medication review and clinical pharmacy” course (4 ECTS credits at the time of the study) should be compulsory for all pharmacy students (*n* = 15).

## 4. Discussion

We studied third-year pharmacy students’ competency in medication reviews and their feedback for further development of medication review training in their curriculum. To our knowledge, this is the first study in which medication review competency is self-assessed by undergraduate pharmacy students using nationally defined competencies as a reference. Previous studies in relation to this topic come mostly from the United States and have focused on assessing the impact of single medication review-related courses on students’ competency [31,32,33,34,35,36,37,38]. Instead of concentrating on one course, our study focused on the competencies fostered in a 3-year pharmacy curriculum that was recently reformed, with one of the added learning outcomes targeting medication reviews. The student segment used in this study was the second graduating class from the reformed curriculum.

As expected, self-assessed medication review competency was higher in knowing the general principles of reviewing medications (Figure 2) and lower in more demanding competence areas related to applied clinical pharmacotherapy, such as deprescribing and following up with medication changes (Figure 3). Advanced medication review competencies are demanding and require clinical experience. The results in self-estimated competencies indicate a lack of experience. The competency deemed weakest by the students’ self-assessment was “applying clinical information from laboratory tests to patient care and knowing which laboratory tests are most important to monitor in each condition and medication”. Moreover, “monitoring and improving medication adherence and self-management” and “knowing current care guidelines and applying them to practice” were self-assessed as rather weak. These competence areas are also considered difficult by practicing pharmacists [28,39]. This may be because patient-care-oriented practice is not routine for practicing community pharmacists in Finland. Nevertheless, a remarkable extension of clinical pharmacy services has recently occurred in hospitals [40]. An increased range of services has become available and is now provided in the clinics and wards, which should be considered in future curriculum development. Students had more confidence in competence areas involving the use of medication risk assessment databases, which are integrated into pharmacy prescription processing systems [39]. These databases help to identify and solve medication-related risks in routine dispensing and assist in providing information to medicine users. The use of these tools is actively taught to students throughout the curriculum. According to Pitkä et al., during the first required 3-month internship period (second study year), students learned to utilize product-related databases to assist in medication counseling [41]. This present study indicates that after their second internship, at the time of graduation as BSc pharmacists, the databases assisting in medication reviews were known and frequently used by the students.

The students’ self-assessment results indicate the same kinds of strengths and weaknesses in medication review competencies that have been previously found in practicing pharmacists in Finland [41] and PharmD students in the United States [33]. A study with United States PharmD students indicated that their self-estimated skills in providing accurate information to patients regarding medication and condition improved from 3.5 to 4.2 (mean score on a 5-point Likert scale, ranging from 1 for “strongly disagree” to 5 for “strongly agree”) during a voluntary medication review-related course. The course consisted of lectures, paper-based cases, and an actual patient encounter.

### 4.1. Strengths and Limitations

This study represents a method to involve students in curricular assessment. To our knowledge, similar studies are scarce concerning fostering medication review competencies. The strength of this study is the high participation rate (93%) among third-year students completing their internship (*n* = 102). The self-assessment instrument seemed to differentiate student responses, offering different assessment results depending on the comprehensiveness of the competency area. Self-assessment results were not yet compared to other cohorts of third-year students, but this will be the goal in future studies. A limitation of this study is that we did not have comparable self-assessment results from the time before the curriculum reform so we could have seen the impact of the reform on learning outcomes. It is also important to keep in mind that the self-assessment results may differ from the actual competency each student holds. According to the “Dunning-Kruger effect”, students with less competency often fail to recognize their incompetency, thus overestimating their skills [42]. It is also possible that students responded in the self-assessment in a way to please those evaluating the students’ assessment, although the students were not given a grade from the self-assessment.

Another limitation of the study is that the comprehensiveness of medication reviews conducted during the internships may have varied between students, from more technical prescription reviews to comprehensive clinical reviews, which may have influenced their competency development and self-assessment. This is because different medication reviews require different competence areas, and thus, students may self-assess their competency as higher or lower depending on the comprehensiveness of the review they conducted. The comprehensiveness of the review (prescription review, medication review, or comprehensive medication review) conducted should have been asked in the self-assessment instrument.

### 4.2. Implications

Study findings suggest that improvements to the curriculum are needed. Although students did not self-assess their competency as very poor in any of the competence areas, some of the areas had lower self-assessments, which indicates that more efforts should be focused on training students in those competency areas, particularly clinical skills in patient care and medication management. Moreover, students had a relatively low mean grade in systematic pharmacology (Table 1), which suggests that improvements are needed in teaching students these skills.

Further education development should include more learning about patient case scenarios. In this study, over a third of the students suggested that their training should contain more medication review assignments as group work, in which learning is patient-centered and self-directed. Learning should take place both at the university and during clinical internship periods at community pharmacies, hospitals, and other work environments. At the University of Helsinki, training has evolved from lecture-based active pharmaceutical ingredient (API)-centered learning to patient-centered cases since 2014 [43]. To assess the outcomes of curriculum development and applied training methods, we are repeating this self-assessment annually with the third-year students.

Future studies will focus on investigating more deeply students’ skills in monitoring the effects of patients’ medications and improving medication adherence and self-management. As the self-assessment is included in the required assignments supporting learning during internships [44], it facilitates future studies to compare competency development longitudinally and to see the potential effects of curriculum changes on medication review competency. Our plan is also to involve preceptors in assessing the medication review competencies of students.

## 5. Conclusions

Most of the third-year pharmacy students self-estimated their medication review competency as good or very good. The highest competence self-assessments related to knowing the general principles of reviewing medications, applying databases to medication risk management, and understanding the importance of medication review in improving medication safety. Further curriculum development efforts concerning fostering medication review competencies should focus on the lowest self-assessed competence areas: monitoring and improving adherence and supporting self-management, making contacts with other health care providers, and interpreting laboratory values considering conditions and medications. Enhanced use of case-based learning would be an effective method for fostering these clinical competencies.

## Figures and Tables

**Figure 1 ijerph-20-05079-f001:**
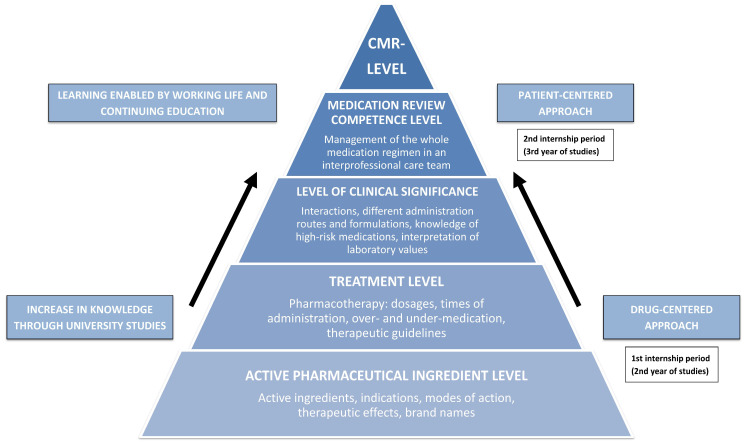
Medication review competence development model [12,19]. CMR: comprehensive medication review [2].

**Figure 4 ijerph-20-05079-f004:**
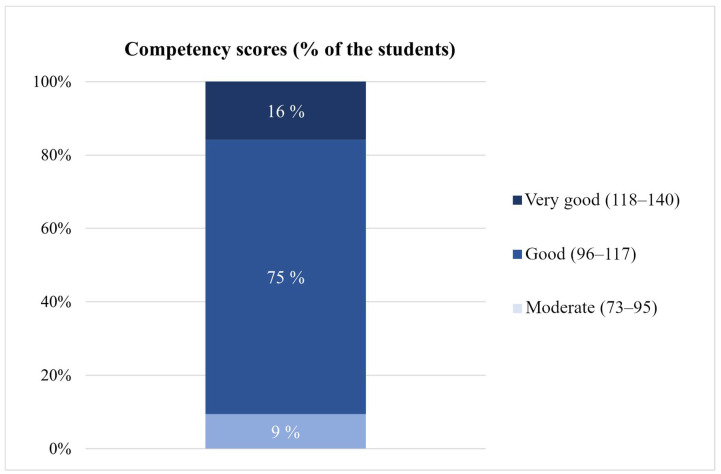
Grading of students’ (*n* = 95) medication review competency on the basis of the summative competency scores (score range 28–140). The competency score was formed as a summative variable from students’ self-assessment.

**Table 1 ijerph-20-05079-t001:** Characteristics of the pharmacy students who self-assessed their medication review competency (*n* = 95). Obligatory courses were graded 1–5, 5 being the highest grade (SD stands for standard deviation).

Variable		
Curriculum, *n* (%)	95 (100)	
Bachelor’s degree	74 (78)	
Master’s degree	21 (22)	
Years studied	Mean	Range
BSc (Pharm) students	3.2	3–6
MSc (Pharm) students	3.0	3–4
Previous degrees, *n* (%)	95 (100)	
No previous degrees	66 (70)	
Bachelor’s or master’s degree, university	12 (13)	
Bachelor’s degree, university of applied sciences	8 (8)	
Vocational qualification	7 (7)	
PhD or equal	2 (2)	
Grades in obligatory courses laying foundation in clinical pharmacy (grade 1–5, 5 being the highest)	Mean grade	SD
Medication Counselling and Pharmacotherapy (4 ECTS credits)	4.3	0.7
Pathology and Nutrition (4 ECTS credits)	3.1	1.2
Systematic Pharmacology (12 ECTS credits)	2.4	1.3
Has the student taken the elective course “Comprehensive medication review and clinical pharmacy” (4 ECTS credits), *n* (%)	95 (100)	
Yes	3 (3)	
No	92 (97)	

## Data Availability

The data presented in this study are available on request from the corresponding author.

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
