# Peer review of "Developing Medication Review Competency in Undergraduate Pharmacy Training: A Self-Assessment by Third-Year Students"

_ijerph, 2023, doi:10.3390/ijerph20065079_

Round 1

Reviewer 1 Report

This is an interesting study about how third year pharmacy students perceive their competency for performing medication reviews in Finland. The writing is clear and I have only a few minor comments to make.

Lines 11 and 37: Is it really "medication history taking" or 'medication-taking history"?

Line 56: Although authors included an Appendix that describes the three levels of comprehensiveness for medication reviews, I would suggest a brief description of the differences between prescription review, medication review and comprehensive medication review in the article text and a link of these three level with figure 1.

Line 97. I was surprised to read that a physician could supervise students' medication reviews. Was this frequent in your sample? Could this have led to different assessments of competencies by these students? I would have like authors discuss this in their paper.

Line 147-148. I do not understand this sentence "Students were written to that responding to the e-form was regarded as an informed consent...". To be clarified.

Presentation of Table 1 could be improved. First, I would present the number before the %. Total should be added to column titles (and this line could be removed). Abbreviations should be defined as table footnotes. I would also replace "Variable" by a more meaningful title.

Lines 168-173. I have noted that some competencies begin with "knowing" while others begin with "mastering" and that those competencies beginning with mastering have lower proportions of "good or very good competency". I would like authors to comment on that in their paper. They should discuss why those competencies need to be mastered and if this might have influenced the results (more difficult than only "knowing")?

Line 228. I think there is a "to" missing..."According to Pitka..."

Line 278. In the discussion authors rightly note that a limitation of their study is that it was a self-assessment of students' competencies. Is it also possible that students might have responded in a way to "please" those reading the evaluations thinking, for instance, it could be noted or important for their grade? I think this possibility should be pointed out by authors in their limitations.

I would also suggest authors to discuss how their study is generalizable to other countries and how results can be apply elsewhere.

Author Response

Dear Reviewer,

Thank you for the valuable comments on our manuscript. We have looked into your suggestions with care and done the best we could to meet your expectations for this manuscript and revised it accordingly. Please see our point-by-point response to your comments below.

Reviewer’s comment: Lines 11 and 37: Is it really "medication history taking" or 'medication-taking history"?

Author’s response: We double checked this and came into the conclusion that it is medication history taking.

Reviewer’s comment: Line 56: Although authors included an Appendix that describes the three levels of comprehensiveness for medication reviews, I would suggest a brief description of the differences between prescription review, medication review and comprehensive medication review in the article text and a link of these three level with figure 1.

Author’s response: Thank you for this suggestion. We have added a brief description of differences between reviews from line 57 onward.

Reviewer’s comment: Line 97. I was surprised to read that a physician could supervise students' medication reviews. Was this frequent in your sample? Could this have led to different assessments of competencies by these students? I would have like authors discuss this in their paper.

Author’s response: Thank you for this suggestion. Unfortunately, we didn’t ask specifically, which profession did the supervisor have. We considered your suggestion, but while we don’t have the information of this, adding this topic to discussion would not bring enough value.

Reviewer’s comment: Line 147-148. I do not understand this sentence "Students were written to that responding to the e-form was regarded as an informed consent...". To be clarified.

Author’s response: Thank you for noticing this unclear sentence. This is now corrected to the manuscript.

Reviewer’s comment: Presentation of Table 1 could be improved. First, I would present the number before the %. Total should be added to column titles (and this line could be removed). Abbreviations should be defined as table footnotes. I would also replace "Variable" by a more meaningful title.

Author’s response: Thank you for this extensive suggestion on the Table 1. Numbers are now presented before percentages. The “total” is now represented in a different way. Abbreviation for SD is now added in the footnotes. We considered your comment on the term “Variable” and looked other published IJERPH articles. The term was in use in other publications as well, so we would argue that it is a meaningful title also in this table.

Reviewer’s comment: Lines 168-173. I have noted that some competencies begin with "knowing" while others begin with "mastering" and that those competencies beginning with mastering have lower proportions of "good or very good competency". I would like authors to comment on that in their paper. They should discuss why those competencies need to be mastered and if this might have influenced the results (more difficult than only "knowing")?

Author’s response: Thank you for this comment. The differences in reviews are added in line 57 onward. The differences in student competency are discussed in the discussion (lines 242-254 and 258-267). We consider this topic to be discussed to the extent that is needed.

Reviewer’s comment: Line 228. I think there is a "to" missing..."According to Pitka..."            

Author’s response: Thank you for noticing this error. This is now corrected.

Reviewer’s comment: Line 278. In the discussion authors rightly note that a limitation of their study is that it was a self-assessment of students' competencies. Is it also possible that students might have responded in a way to "please" those reading the evaluations thinking, for instance, it could be noted or important for their grade? I think this possibility should be pointed out by authors in their limitations.

Author’s response: Thank you for this comment. It is possible, and the manuscript has been revised accordingly (lines 323-325).

Reviewer’s comment: I would also suggest authors to discuss how their study is generalizable to other countries and how results can be apply elsewhere.

Author’s response: Thank you for this suggestion. We would think that the results are not easily generalizable to other countries, as they are highly dependable on the curriculum, which can be quite complex and difficult to compare. But, the study protocol can be used in different countries. The manuscript has been revised accordingly (lines 305-310).

Kind regards,

Aleksi Westerholm, MSc (Pharm) (corresponding author)

Katja Leiman MSc (Pharm), PhD Student; Annika Kiiski, MSc (Pharm), PhD Student; Marika Pohjanoksa-Mäntylä, MSc (Pharm), PhD; Anita Mistry, BSc (Pharm); Marja Airaksinen, MSc (Pharm), PhD, Prof

Reviewer 2 Report

The authors appear to have done a cross sectional survey of student’s self-assessed competency after a medication review assessment.

While I believe it is important to assess the impact of educational innovations I do not believe that the authors have done this. There are also significant reporting issues in this manuscript (e.g. unclear what the educational changes have been, etc should others want to implement it is impossible to do so based on what is reported, not reporting the qualitative findings etc) – however, I will focus on study design as I believe that this is most important.

The authors had an opportunity to find out whether the students had genuine competency as assessed by the supervisors and see if this correlated well with student self-assessed competency yet failed to do so. There is no control group (historical or otherwise) to help measure the impact the new pharmacy education curriculum has had on students medication review competencies. Surely the authors have access to previous year’s marks at national level exams – they could compare this with current graduates in an interrupted time series approach. If on the other hand the authors wanted to understand the views and experiences of students going through the curriculum, in depth interviews with current and past graduates would have provided more insight into the impacts of the educational reform.

While doing this kind of work may be important for internal quality assurance/improvement and is valuable data for any grant applications, it is not clear from the manuscript what this study adds to the broader educational literature.

Author Response

Dear Reviewer,

Thank you for the valuable comments on our manuscript. We have looked into your suggestions with care and done the best we could to meet your expectations for this manuscript and revised it accordingly. Please see our point-by-point response to your comments below.

Reviewer’s comment: The authors appear to have done a cross sectional survey of student’s self-assessed competency after a medication review assessment. While I believe it is important to assess the impact of educational innovations I do not believe that the authors have done this.

Author’s response: Thank you for this comment. The purpose of this study was not to assess the impact of an educational innovation, but to assess students’ competency in medication reviews and their feedback for further development of medication review training in their curriculum.

Reviewer’s comment: There are also significant reporting issues in this manuscript (e.g. unclear what the educational changes have been, etc should others want to implement it is impossible to do so based on what is reported, not reporting the qualitative findings etc) – however, I will focus on study design as I believe that this is most important.

Author’s response: Thank you for this comment. The educational changes are described in the lines 98-104. It would indeed be difficult to implement our curriculum changes to another country since the content of even a single course can be quite complex and thus difficult to copy to another university. This would not be possible to describe in a single article. The purpose of this manuscript was not to explain the curriculum changes in detail, but rather assess third-year pharmacy students’ competency in medication reviews and their feedback for further development of medication review training in their curriculum. Also, there are qualitative findings in this manuscript, since there was possibility for the students to answer in open questions, results are represented on line 224-229.

Reviewer’s comment: The authors had an opportunity to find out whether the students had genuine competency as assessed by the supervisors and see if this correlated well with student self-assessed competency yet failed to do so.

Author’s response: Thank you for this comment. We are aware of this limitation of the study and would agree that supervisors’ assessment would bring extra value. Unfortunately, by the time doing the research, we did not have this information, and it is discussed in lines 358-359.

Reviewer’s comment: There is no control group (historical or otherwise) to help measure the impact the new pharmacy education curriculum has had on students medication review competencies.

Author’s response: Thank you for this comment. We are aware of this limitation of the study and would agree that a control group would bring extra value. Unfortunately, by the time doing the research, we did not have this information, and this is discussed in lines 319-321 and in lines 350-352.

Reviewer’s comment: Surely the authors have access to previous year’s marks at national level exams – they could compare this with current graduates in an interrupted time series approach. If on the other hand the authors wanted to understand the views and experiences of students going through the curriculum, in depth interviews with current and past graduates would have provided more insight into the impacts of the educational reform.

Author’s response: Thank you for this comment. The comparison to previous year’s grades would not present the desired effects of the curriculum reform. Reform was supposed to enhance competency in medication reviews, which was assessed in this study, as it is described in lines 90-95. Also, this was not a qualitative study, but there were open questions for students to tell us how to further develop the curriculum in order to improve medication review competency, as it is described in lines 224-229.

Reviewer’s comment: While doing this kind of work may be important for internal quality assurance/improvement and is valuable data for any grant applications, it is not clear from the manuscript what this study adds to the broader educational literature.

Author’s response: Thank you for this comment. This study’s novelty and added value to the educational literature are now described more clearly in lines 305-311.

Kind regards,

Aleksi Westerholm, MSc (Pharm) (corresponding author)

Katja Leiman MSc (Pharm), PhD Student; Annika Kiiski, MSc (Pharm), PhD Student; Marika Pohjanoksa-Mäntylä, MSc (Pharm), PhD; Anita Mistry, BSc (Pharm); Marja Airaksinen, MSc (Pharm), PhD, Prof

Reviewer 3 Report

I recommend the manuscript titled " Developing medication review competency in undergraduate pharmacy training: A self-assessment by third-year students " to be accepted after minor revision done, therefore I propose:

1.      Introduction section: I recommend to define more precisely the aim of the work, to formulate it in such a way as to correspond to the Conclusions.

2.      The using of the Likert scale may introduce some inaccuracies, in this case due to the fact that respondents will try to make a favorable impression by answering unfairly. Please evaluate the impact on the results of the study.

Author Response

Dear Reviewer,

Thank you for the valuable comments on our manuscript. The manuscript has been revised accordingly. Please see our point-by-point response to your comments below.

Reviewer’s comment: I recommend the manuscript titled " Developing medication review competency in undergraduate pharmacy training: A self-assessment by third-year students " to be accepted after minor revision done.

Author’s response: Thank you for this amiable comment. We have looked into your suggestions with care and done the best we could to meet your expectations for this manuscript.

Reviewer’s comment: Introduction section: I recommend to define more precisely the aim of the work, to formulate it in such a way as to correspond to the Conclusions.

Author’s response: Thank you for this suggestion. The aim of the work is presented in lines 108-110. We double checked, that the topics introduced in the introduction are met in the conclusions, lines 361-370. We would argue that the aims are precisely defined in the present manuscript.

Reviewer’s comment: The using of the Likert scale may introduce some inaccuracies, in this case due to the fact that respondents will try to make a favorable impression by answering unfairly. Please evaluate the impact on the results of the study.

Author’s response: Thank you for this suggestion. This is now discussed in lines 324-326.

Kind regards,

Aleksi Westerholm, MSc (Pharm) (corresponding author)

Katja Leiman MSc (Pharm), PhD Student; Annika Kiiski, MSc (Pharm), PhD Student; Marika Pohjanoksa-Mäntylä, MSc (Pharm), PhD; Anita Mistry, BSc (Pharm); Marja Airaksinen, MSc (Pharm), PhD, Prof

Author Response

Dear Reviewer,

Thank you for the valuable comments on our manuscript. We have looked into your suggestions with care and done the best we could to meet your expectations for this manuscript and revised it accordingly. Please see our point-by-point response to your comments below.

Reviewer’s comment: <to research> - “to study” (line 12)

Author’s response: Thank you for this suggestion. This is now changed to “to investigate”, since the sentence would otherwise be “…objectives of this study were to study…”

Reviewer’s comment: <of their second 3-month internship period in a community pharmacy in 2017-2018.> in most countries in Europe the way of becoming a pharmacist is different! (lines 15-16)

Author’s response: Thank you for this comment. The two-tier education system in Finland is discussed in lines 96-98.

Reviewer’s comment: <The self-assessment was carried out via e-Form> - please add whether you made this Form or that you used an existing one (line 18)

Author’s response: Thank you for this suggestion. This is now added in lines 18-19.

Reviewer’s comment: <Students (n=95, participation rate 93%)> - how do you know about this participation rate? Or is it theory (did you use the number of students who ought to be in school)?? (line 20)

Author’s response: Thank you for this comment. This is discussed later in the manuscript (lines 171-172), as you noticed in your latter comment.

Reviewer’s comment: <Clyne et al> - . (et is not an abbreviation; al. is [from alii] so a dot) (line 55)

Author’s response: Thank you for noticing this error. This is now revised accordingly.

Reviewer’s comment: Why is this educational intervention only in the bachelor phase, and not in the masters as well? Students might have forgotten what they once learned about the patient (5-6 years ago) and return to the drugs?

Author’s response: Thank you for this comment. In Finland, most of the pharmacy students only study the bachelor’s degree, and after that are eligible to work in a community pharmacy. That is why this intervention is only at the bachelor’s degree level. The two-tier education system in Finland is discussed in lines 96-98.

Reviewer’s comment: <The student were instructed> - either: “The student was instructed” or “The students were instructed” (line 122)

Author’s response: Thank you for noticing this error. This is now revised accordingly.

Reviewer’s comment: Please rewrite this section: First, we … . Then, we … . Next, we … . Finally, we … . The readership easier grabs what you did and in which order the Results will be shown (lines 153-155)

Author’s response: Thank you for this suggestion. This is now revised accordingly.

Reviewer’s comment: Now I understand the participation rate! (line 171)

Author’s response: Great!

Reviewer’s comment: Please keep in mind the following structure for writing a Discussion:

para1 start with repeating the research question + answer this without any comments or without any interpretation.
Eg start with: We studied the third-year pharmacy students’ competency in medication reviews and their feedback for further development of medication review training in their curriculum.
Para2,3,# start a new para, 1 topic per para, and start this para with one of your findings which then defines the content of the para. Relate your finding to earlier published references.

Author’s response: Thank you for this extensive suggestion on the discussion. The discussions-section is now revised with 1 topic per paragraph.

Reviewer’s comment: Strengths and limitations
Please add this heading
Mention strengths clearly. Limitations are those issues which might bias your findings (are there sources of bias, like information bias or selection bias and how did you handle confounders?).

Author’s response: Thank you for this suggestion. This is now revised accordingly (lines 304-335).

Reviewer’s comment: Implications
Please add a paragraph on what your findings mean for practice / policy, and what they mean for future research.

Author’s response: Thank you for this suggestion. This is now revised accordingly (lines 336-359).

Reviewer’s comment: 41 Schepel L, Aronpuro K, Kvarnström K, Holmström A, Lehtonen L, Lapatto-Reiniluoto O, Laaksonen R, Carlsson K, Airaksinen M: Strategies for improving medication safety in hospitals: Evolution of clinical pharmacy services. Res Soc Adm Pharm (lines 481-483)

Should not be underlined

Author’s response: Thank you for noticing this error. The reference is now revised accordingly.

Reviewer’s comment: 44. Faculty of Pharmacy, University of Helsinki. Työ opiksi! (line 498)

[add English translation of Työ opiksi!]

Author’s response: Thank you for your suggestion. The reference is now revised accordingly.

Kind regards,

Aleksi Westerholm, MSc (Pharm) (corresponding author)

Katja Leiman MSc (Pharm), PhD Student; Annika Kiiski, MSc (Pharm), PhD Student; Marika Pohjanoksa-Mäntylä, MSc (Pharm), PhD; Anita Mistry, BSc (Pharm); Marja Airaksinen, MSc (Pharm), PhD, Prof